# AN EXHAUSTIVE ANALYSIS OF LAZY VS. EAGER LEARNING METHODS FOR REAL-ESTATE PROPERTY INVESTMENT*

## ABSTRACT

Accurate rent prediction in real estate investment can help in generating capital gains and guaranty a financial success. In this paper, we carry out a comprehensive analysis and study of eleven machine learning algorithms for rent prediction, including Linear Regression, Multilayer Perceptron, Random Forest, KNN, ML-KNN, Locally Weighted Learning, SMO, SVM, J48, lazy Decision Tree (i.e., lazy DT), and KStar algorithms. Our contribution in this paper is twofold: (1) We present a comprehensive analysis of internal and external attributes of a real-estate housing dataset and their correlation with rental prices. (2) We use rental prediction as a platform to study and compare the performance of eager vs. lazy machine learning methods using myriad of ML algorithms. We train our rent prediction models using a Zillow data set of 4K real estate properties in Virginia State of the US, including three house types of single-family, townhouse, and condo. Each data instance in the dataset has 21 internal attributes (e.g., area space, price, number of bed/bath, rent, school rating, so forth). In addition to Zillow data, external attributes like walk/transit score, and crime rate are collected from online data sources. A subset of the collected features - determined by the PCA technique-are selected to tune the parameters of the prediction models. We employ a hierarchical clustering approach to cluster the data based on two factors of house type, and average rent estimate of zip codes. We evaluate and compare the efficacy of the tuned prediction models based on two metrics of R-squared and Mean Absolute Error, applied on unseen data. Based on our study, lazy models like KStar lead to higher accuracy and lower prediction error compared to eager methods like J48 and LR. However, it is not necessarily found to be an overarching conclusion drawn from the comparison between all the lazy and eager methods in this work.

## 1 INTRODUCTION

Real estate rent prediction has a key role in calculating the Rate of Return- a measure for evaluating the performance of an investment in the housing market. Rate-of-Return can measure the quality of a real estate investment over a time-period and Net Present Value (NPV) is one of its important factors (Investopedia)[1] . In real-estate NPV defines the profitability of the investment based on rate of return. The NPV yields an accurate insight to real estate investors on whether they achieve a satisfactory rate of return within a certain period of time. Equation 1 shows how to calculate NPV[2]:

$$NPV = \sum_{n=0}^{N} \frac{CF_n}{(1+r)^n} \tag{1}$$

In equation 1, *CF* is the cash flow generated from a rental property for each period *n* in the holding period *N* (i.e., the time period of an investment), and r is the desired investor rate of return. Based on equation 1, one of the important factors in evaluating the NPV of a real estate property investment is cash flow (CF) that is calculated based on equation 2 for a 12-month period:

---

[1]`https://www.investopedia.com/terms/f/futurevalue.asp`
[2] `https://www.propertymetrics.com/blog/2015/06/11-/what-is-npv/`

$$CF = 11\rho - (12\mu + \tau + \epsilon + \iota) \tag{2}$$

where $\rho$ is the rent income (for simplicity, the Vacancy rate analysis is not discussed here, and the assumption is based on 11-months rent income per year), $\mu$ is the house mortgage, $\tau$ is the annual house tax, $\epsilon$ is the annual house expenses, and $\iota$ is the mortgage house insurance. In equation 2, $\rho$ is only factor that has a positive effect on cash flow. Therefore, it is very essential to provide an accurate rent prediction method. For many people, house is an invaluable asset. Therefore, having a safe investment is a significant task. Proper rental property investments can lead to a successful and profitable Rate of Return over time. However, such ventures can be very risky due to miscalculation or inaccuracy of algorithms used in rent prediction. Applying machine learning algorithms to perform house rent prediction is not a novel trend. Lambert & Greenland (2015) investigates eager learning methods like MLP, and bagging REP trees to estimate the rental rate for both the land-owners and students interested in renting a place close to a university campus. The coverage area of the training set is limited to three distant zip codes surrounding a university campus. The input features entertained in this work include proximity to university campus, apartment appliances and dimensions, the length of the apartment contract, and the date of the residence's constructions. The study reports bagging REP trees as the best rent prediction algorithm. However, the proposed global learning-based solution can generate a biased model due to the skewed data set, all located surrounding a university campus. In the previous studies, the prediction models for real estate rent/price prediction are very generic and they don't differentiate according to the house type and/or zip code (Limsombunchai, 2004; Yu & Wu, 2016). Figure 1 shows zip code-wise variation of the rent behavior, even for the real estate properties which are in the same state/city or a close geo-spatial proximity from each other. For instance, the zip codes 22066 and 20190 are neighbors but they show a very different behavior in terms of the average rent price.

| 22066 | 22125 | 22044 | 22079 | 20112 | 20137 |
|-------|-------|-------|-------|-------|-------|
| 22027 | 20117 | 20197 | 20152 | 20194 | |
| 22102 | 22181 | 20169 | 20181 | | |
| | | 20176 | 22030 | 20136 | |
| 22124 | 22043 | | | 20187 | |
| 20124 | 20175 | 20143 | 20105 | 22026 | |
| | | 20148 | 20180 | 20121 | |
| 22180 | 22039 | 20132 | 22031 | 20190 | |
| 22101 | 22189 | 20184 | 20158 | 20166 | |

Figure 1: Average rent prices for real-estate properties based on zip code in the state of VA. The dark blue shows higher rent prices compared to the bright blue color which indicates lower rent prices.

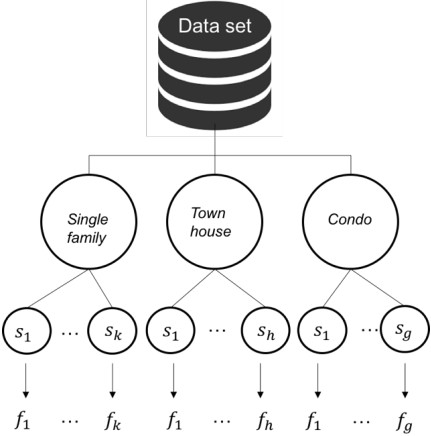

Figure 2: Two-layer clustering technique, first according to house type, and then based on average rent price.

In addition, statistical data shows that house type parameter affects the rent price due to internal factors like area space, number of bed/bathrooms, HOA fee, community factors, so forth (see Figure 3). The average rent price for a zip code depends on internal factors (like house type, number of bed/bath, price, area space, other) and external factors. In fact, external factors like walk/transit score, crime rate, and school ratings corresponding to a zip code impact the price of rent and are deal-breakers for many real estate investors (US News)[3]. In this paper, internal and external house properties are entertained. Walk score indicates the errands that can be accomplished on foot or those that require a car to nearby amenities. Transit score indicates the connectivity (i.e., proximity

---

[3]https://realestate.usnews.com/real-estate/articles/how-homicide-affects-home-values

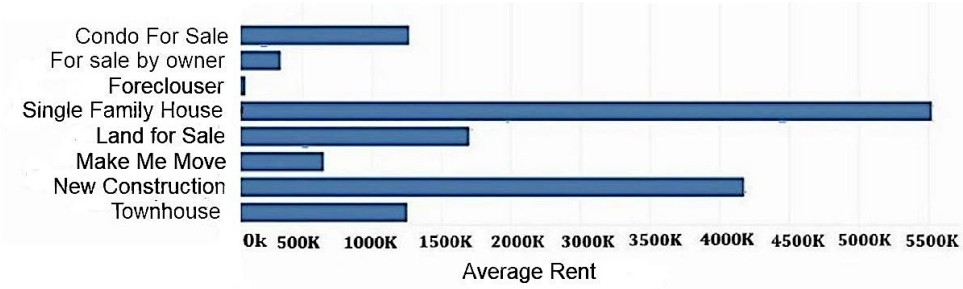

Figure 3: Rent distribution for different house types for multiple zip codes.

to metro), access to jobs, and frequency of service. Crime score indicates the rate of violent and non-violent incidents related to a zip code (FBI) [4] .

In the past studies, the impact of eager learning methods for real estate rent/price prediction has been investigated (Lambert & Greenland, 2015). Eager methods require less space in comparison with lazy algorithms. However, in the real estate rent prediction domain, we are not dealing with streaming data, and so data volume is not a critical issue. In general, unlike eager learning methods, lazy learning (or instance learning) techniques aim at finding the local optimal solutions for each test instance. Kohavi et al. (1996) and Homayouni et al. (2010) store the training instances and delay the generalization until a new instance arrives. Another work carried out by Galván et al. (2011), compares a limited number of memory-based learning vs neural network methods, and report the superiority of memory-based method over Neural Network approach using multiple data sets of UCI machine learning data set repository, including Iris, Diabetes, Sonar, Vehicle, car, and balance. In this study, the two lazy learning algorithms, namely Integer Part and Atomic Radios outperformed eager learning algorithms, namely SMO and Naive Bayes algorithm. Our work is inspired by these research explorations.

We collected a Zillow data set of 4K real estate properties in Virginia State. In addition, transit/walk score, and crime rate are collected from information sources like alltransit.cnt.org, walkscore.com, and crimereports.com respectively. The transit parameters entertained in the data collected from All-Transit data source clearly indicates the proximity to metro as a significant parameter in determining the transit score of a location.

The data set consists of three house types: town-house, single family, and condo. This study is motivated by the need to build models with respect to house type and zip code. To deal with the sparse (i.e., thin) data in every zip code, we divided the data set according to house type, and then applied K-means clustering to generate subsets of instances within zip codes with similar average rent prices as illustrated in Figure 2. The clustering method uses the similarity measure of average-rent to compute the distance between the data points. The data samples in each cluster is later used to train a rent prediction model. In this work, we study the impact of several machine learning methods on this data set by performing a comparative analysis of lazy vs. eager learning methods. We report the empirical results to compare the performance of lazy vs. eager learning algorithms in real-estate rent prediction for each house type and a subset of zip codes with similar average rent prices. We examine the performance of Linear Regression (LR), Sequential Minimal Optimization (SMO), Multilayer Perceptron (MLP), J48, SVM, and Random Forest (FR) algorithms (eager/globally-based learning) against K-Nearest Neighbors (KNN), Multi-Label KNN (ML-KNN), lazy Decision Tree (Lazy-DT), locally weighted learning (LWL) and KStar (K*) algorithms (lazy/memory-based learning/instance-based), using two performance evaluation metrics: i) R-squared and ii) Mean Absolute Error (MAE). The target variable is the rent price and the evaluation metrics show the variance between the predicted target variable and the actual rent price. Our rent prediction algorithm uses a salient subset of data set attributes, which is determined during feature selection using Principal Component Analysis technique (PCA). PCA technique filters out unwanted features based on each house type (i.e., single family, town house, and condo), and independent of the learning algorithm applied on the data. For

---

[4] https://ucr.fbi.gov/crime-in-the-u.s/2011/crime-in-the-u.s.-2011/violent-crime/violent-crime

imputation, we removed the observations with many missing attributes as the proportion of these instances to the entire data set was less than 3%. The remainder of this paper is structured as follows: In section 2 related work is discussed. We describe the data set used in this paper for analysis in section 3. Section 4 describes our methodological framework including data preprocessing, data exploration and feature selection, building prediction models, and model evaluation. In section 5, experiments and results are discussed. Finally, section 6 gives the conclusion.

## 2    RELATED WORK

Real estate rent/price prediction using machine learning techniques has already been studied in several works (Galván et al., 2011; Limsombunchai, 2004; Khamis & Kamarudin, 2014; Basu & Thibodeau, 1998; Webb, 2011; Kuntz & Helbich, 2014). In Wang et al. (2014), PSO-SVM algorithm is used for real-estate price prediction. Liu (2013) uses spatiotemporal dependencies between housing transactions to predict future house prices. This approach is limited by spatial autocorrelation, since the degree of similarity between observations is not solely based on the distance separating them. Some of the previous work focus on hedonic price models as a method of estimating the demand and value in the housing market and determination of house prices (Önder et al., 2004; Ozus et al., 2007). In these studies, rather than internal and external house features, economic submarkets are used in the prediction model which are defined in terms of the characteristics of neighborhoods or census units. Limsombunchai (2004) uses a sample size of 200 houses of all house types in New Zealandto train a hedonic price and an artificial neural network (ANN) model, and shows that the eager method ANN outperforms the hedonic model. The problem with the hedonic approach is disregarding the differences between the properties in the same geographical area. Park & Bae (2015) determine the house sales trends of the US housing market subject to the Standard & Poor's Case-Shiller home price indices and OFHEO which is the housing price index of the Office of Federal Housing Enterprise Oversight. The authors used and compared the classification accuracy of four methods C4.5, RIPPER, Naïve Bayesian, and AdaBoost, where RIPPER algorithm outperformed others. We are now at the age where people in different fields are hacking their way into machine learning. Machine learning techniques have become available as commodities which can be used to perform prediction and classification tasks in various domains like real-estate rent/price prediction. Khamis & Kamarudin (2014) compared the efficacy of the eager learning method Neural Network (NN) against the hedonic model Multiple-Linear Regression (MLR), and showed that NN outperforms MLR. However, Galván et al. (2011) reports the superiority of lazy learning methods over NN. According to Webb (2011), eager learning methods can sometimes lead to suboptimal predictions because of deriving a single model that seeks to minimize the average error over the entire data set, whereas lazy learning can help improve prediction accuracy. While our study is inspired by Galván et al. (2011) and Webb (2011), we take our analysis to the next level, by comparing the impact of eager and lazy learning algorithms in the prediction accuracy of the generated models with respect to each house type and a subset of zip codes with similar average rent prices. We use a two-layer clustering technique, and a subset of internal and external real-estate property factors.

## 3    DATA SET DESCRIPTION

There are variety of housing data sources in the real estate market. Zillow API delivers home details including historical data on sales prices, year of sale, tax information, number of bed/baths, so forth, for the US. In fact, Zillow is tied to various sources like real estate agents, homeowners, tax assessors, public records, and Multiple Listing Service (MLS). Normally, rent prices in real-estate housing do not change abruptly within a very short time window. For example, rent price of a real estate property is not subject to change every day. Hence, there is no need for frequently updating the rent price in the dataset. We analyzed the real-estate rent prices in different zip codes provided by Zillow, and did not find any drastic changes within a period shorter than 4 months, which suggests that the listed rent prices are reasonably reliable. We used the Zillow API to collect a data set of residential housing data for the state of Virginia. The size of this data set contains about 4000 housing property records (including townhouse, single-family, and condo) with 21 attributes. In addition, external attributes, namely walk score, transit score, and crime rate are collected. Figure 4 shows the crime rate for violent and non-violent crime incidents in different zip codes. The description of walk score is illustrated in Table 1.

Table 1: Categorical Description of Walk Score (www.walkscore.com).

| Walk Score | Description |
|---|---|
| 90-100 | Walker's Paradise: daily errands do not require a car. |
| 70-89 | Very Walkable: most errands can happen on foot. |
| 50-69 | Somewhat walkable: some errands can happen on foot. |
| 25-49 | Car-Dependent: most errands require a car. |
| 0-24 | Car-Dependent: almost all errands require a car. |

| zipcode | Population | Incidents | ViolentInc |
|---|---|---|---|
| 20105 | 15,021 | 91 | 9 |
| 20106 | 5,187 | 0 | 0 |
| 20109 | 37,332 | 17 | 3 |
| 20110 | 48,019 | 3 | 1 |
| 20111 | 34,000 | 4 | 1 |
| 20112 | 26,867 | 0 | 0 |
| 20115 | 6,551 | 0 | 0 |
| 20117 | 2,530 | 1 | 1 |
| 20119 | 4,345 | 0 | 0 |
| 20120 | 41,180 | 185 | 57 |

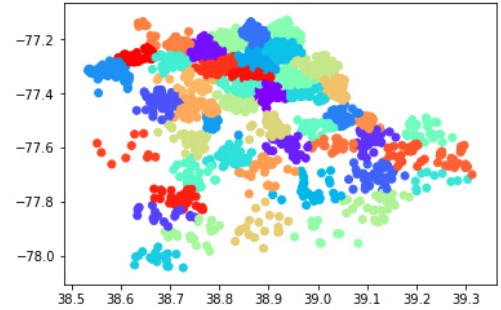

Figure 4: Crime score in the zip code level obtained from www. crimereports.com, including violent and non-violent crimes.

Figure 5: K-means clustering of transit score data generated 48 clusters with maximum distance 6 miles.

We collected transit score data from All Transit source[5] : their dataset is collected from 824 agencies, and it includes 662K stop locations and 13K routs. Transit and walk scores are collected per household, while crime rate is obtained for each zip code , normalized by the number of people living in that area using Selenium tool with Python. Crime score data was normalized using Dickson method (Dickson, 2014) indicated by equation 3:

$$Incident_{norm} = \frac{CrimeIncidents}{population} * 100,000 \tag{3}$$

We obtained zip code-wise population by collecting data from www.moving.com. The zip code-wise correlation between normalized crime rate and population suggests that crime rate is influenced by both geographic areas (i.e., zip codes) and population.

## 4 METHODOLOGY

### 4.1 DATA PREPROCESSING

One of the rudimentary principles in calibration of machine learning models when dealing with a biased data is to re-sample the data to balance them (Sanjuán, 2012). Based on our assessment of the data set, some of the geographic areas have much higher densities compared to other areas. To normalize the data, we re-sampled the data in zip codes with higher house prices due to their crowded density relative to the zip codes with lower house prices. For imputing the missing values of external attributes, we used K-means clustering and KNN. Figure 5 shows the result of this clustering to impute transit score data. The distances between data points is calculated with respect to each cluster centroid. To reduce the dimensionality of the data set and enhance the generalization of the model, we perform feature selection by applying PCA to all 21 attributes of the data set. However, before applying PCA, attributes are normalized based on Min-Max Normalization.

---

[5]https://alltransit.cnt.org

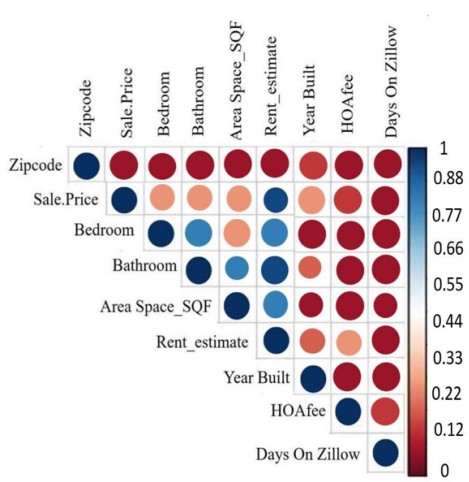

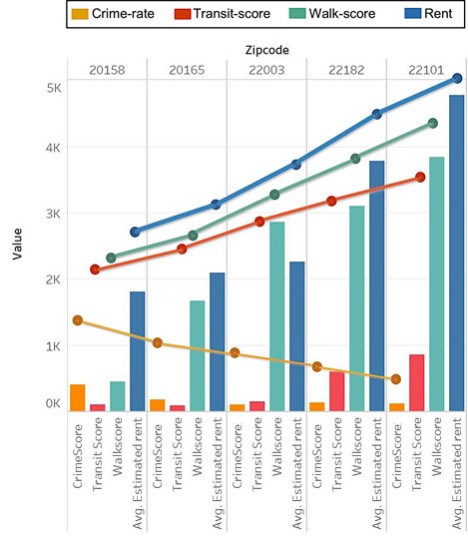

Figure 6: Correlation matrix of VA data set for 3 house types.

Figure 7: Dependency between average rent price across multiple zip codes and urban planning parameters (external attributes) of walk-score, transit-score, and crime-rate.

## 4.2 Data Exploration

We analyzed the correlations between various variables of the data set. Discovering co-linearity between the data set variables and the target variable yields valuable insights about the dependent variables that affect the rent price. Figure 6 illustrates the co-linearity between the data set attributes in the VA data set; since zip code is a nominal attribute, there is no col-linearity between zip code and rent price. Also, there is a strong dependency between rent and internal attributes like number of bed/bathrooms, area space, year, and sale price. Figure 7 illustrates the correlation between average rent price and urban planning parameters of walk/transit-score and crime-rate. The trend in Figure 7 indicates a positive correlation between average rent and walk/transit score, and a negative correlation between the average rent and crime rate across multiple zip codes.

Based on the plots in Figure 1 and 3, we hypothesize that there is a rent prediction model for every house type within a zip code/similar zip codes. To test this hypothesis, we carry out the following analysis: prior to attribute selection, to obtain a suitable representation of the data set, we apply PCA (principle component analysis) to the 21 data set attributes. The attributes consist of ZipID (a unique id for each house in the Zillow API), Number of bed/baths, floor size (the area of the house based on SQF), latitude and longitude (geographical location of each house), year built (the year of house construction), status (house type), zip code, house features (facilities in a house described by owner), estimated rent (basic amount of rent price for each house used as a class label in the prediction task), so forth. Since zip code and house type are nominal attributes, there is no collinearity between the rent price and these two attributes. In addition, the results provided by the PCA technique suggests that zip code and house type attributes should not be included in the subset of attributes used for training the models. Rather, they should be used for clustering the data set. This is further explained in subsection 4.4.

## 4.3 Feature Selection

To identify important attributes, we apply PCA (principle component analysis) - which is a well-known and studied method- on three subsets of data samples, each subset covering a different house type across all zip codes in the state of VA. This is illustrated in Figure 8. Table 2 indicates the attributes selected based on the house type, which is further explained here: unlike town house and

condo, features like HOA fee, walk/transit score show a very low variance for single family instances. The higher variance of transit score, especially for condos explains the outpacing of median appreciation rates of condos compared to single family detached-houses in large metropolitan areas (Harney, 2017). Number of bedrooms is found to be an important feature only when house type is single-family or town-house. Next, unlike town-house instances, average school rating is discovered to be an important feature for both single-family and condo instances. This can be explained due to sparsity of school rating for town-house instances in our data set. In our future work, we will employ data mining techniques to obtain this information for town-house instances. We validated the abovementioned strategy by training our models based on a data set including all house types, and then based on each house type. We discovered that the latter approach leads to relatively higher accuracy and lower prediction error.

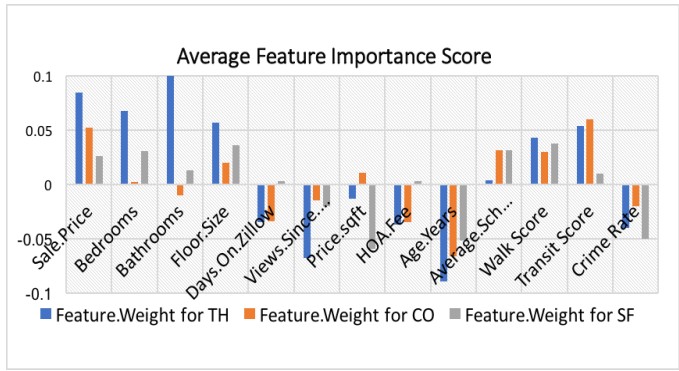

Figure 8: The PCA plot shown separately for townhouse (TH), condo (CO), and single family (SF) instances in all VA zip codes. The vertical axis shows the coefficients of the dominant principal component.

Table 2: Selected Features determined by PCA technique for 3 house types ('SF' for Single Family, 'TH' for Town House, and 'CO' for Condo.

| CO | TH | SF | Attributes |
|:---:|:---:|:---:|:---:|
| ✓ | ✓ | ✓ | price |
|  | ✓ | ✓ | bed/path |
| ✓ | ✓ | ✓ | area |
| ✓ | ✓ | ✓ | views |
| ✓ | ✓ | ✓ | price per SQFT |
| ✓ | ✓ | ✓ | year |
| ✓ |  | ✓ | school rating |
| ✓ | ✓ |  | days-on-zillow |
| ✓ | ✓ |  | HOA |
| ✓ | ✓ |  | walk/transite score |
| ✓ | ✓ | ✓ | crime rate |

## 4.4 DATA CLUSTERING

Based on the statistical analysis in Figure 1, 3, and 8, first, we clustered the housing data set based on house type and zip code attributes, to eventually learn a model for each cluster. However, we observed that the some of the clusters are very sparse with the number of instances below 100, which could immensely affect the ability to train the prediction models (Adomavicius & Zhang, 2012; Li et al., 2016). To increase the density of the training samples and facilitate the accuracy of prediction models at the same time, we carried out a different strategy which is illustrated in Figure 2, described as follows: first, we divided the data set into three groups based on the house type attribute. We refer to these groups as status-clusters. Next, we calculated the average rent for every zip code in each status-cluster. Furthermore, we applied K-means clustering to cluster

the content of each status-cluster based on the average-rent. Using this clustering technique, we increased the density of the training samples.

### 4.4.1 PROBLEM FORMULATION

Given a $status - cluster_j, i = (sf, th, co)$ where sf=single-family, th= town-house, and co=condo with a set of observations $(o_1, o_2, \cdots, o_m)$ where each observation is a $d_i$ dimensional vector, K-means clustering partitions the observations into $n(n = k, h, gs.t., n \ll m)$ sets $S$.

Clustering the data points according average-rent implies organizing instances of similar zip codes inside the same cluster, e.g., cluster $s_j$. The next task is to train a model for each cluster $s_j$ (see Figure 2): $s_j \rightarrow Learner \rightarrow f_j$ , such that $f_j$ is a rent prediction model.

### 4.5 BUILDING PREDICTION MODELS

In this study, as illustrated in Figure 2, we build rent prediction models with respect to each house type and a subset of zip codes with similar average rent prices, using six eager and five lazy learning algorithms. We use WEKA with a customized setting to carry out the implementation (Hall et al., 2009). During the implementation, the dataset is split by 70:30 into train and test sets.

### 4.5.1 DATA PARTITIONING

In our experiments, we use 10-Fold Cross-Validation to partition the training data set into 10 equal parts. During each round of 10 iterations, we repeat the prediction by using one of the 10 parts as test data and the other 9 parts as training data to create a prediction model. Next, we select the model with the best accuracy. We then evaluate the rent prediction model on the test data set that covers 30% of the entire Virginia housing data set collected from Zillow website. We select a subset of well-known eager and lazy learning algorithms described below:

- MLP or Artificial Neural Network ANN with a single hidden layer is an eager learning algorithm and that can approximate nonlinear functions.

- LR is a lazy learning method used in prediction and estimation of real values (like rent, price of houses, so forth). LR can very valuable for business decisions.

- J48 is a variant of C4.5 algorithm. Unlike other lazy trees, J48 does not postpone the prediction decision until completing the construction of the tree.

- SVM is a machine learning method that is used for both classification and regression. A version of SVM used for regression is called Support Vector Regression.

- SMO can be implemented as a lazy or eager version.

- RF is an ensemble learning method which uses several decision trees at the training time, and the output is the mean prediction of the decision trees. Since the algorithm brings extra randomness into the model by searching for the best feature among a random subset of features, it generally results in a model with high accuracy.

- LWL is used to build models that are not learnt from the entire data set. The model is learnt through selected patterns which are created based on the query received. LWL is suitable to generate accurate model when we have plenty of data.

- KStar is an instance-based learner. It has a good performance in the presence of noise and on data sets with imbalanced attributes. KStar is based on the concept of clustering and functions with entropic distance to find the similar instances (Hernández & Dayana, 2015; p.Y & H.B, 2017).

- Lazy-DT is a learning method that for each test instance, builds the best decision tree. It's important advantage is avoiding decision tree problems like fragmentation and replication (Kohavi et al., 1996).

- ML-KNN algorithm is derived from KNN. Based on experiments involving three real-world multi-label learning problems, ML-KNN achieves higher performance compared to some of the popular multi-label learning methods (Zhang & Zhou, 2007).

- KNN is an example of instance-based learning method, which can be used for prediction and estimation.

The important attributes determined during feature selection are used as input of the above-mentioned models to predict the target variable *rent price*.

## 4.6 MODEL EVALUATION

The key comparison measure used for regression analysis and model evaluation described in this section uses two different measures: i) Mean Absolute Error (MAE) and ii) R-squared(R2). MAE measures the accuracy of the prediction models over the test data set. R-squared (or the coefficient of determination) is a quadratic statistical scoring rule which shows how close the actual target data are to the fitted regression line. R-squared is used to show the variance between the predicted target variable and the actual rent price. Hence, the lower MAE and the higher R2, the better our model fits the data. Based on these evaluation metrics, we calculate and compare the efficacy of the produced rent prediction models for eleven machine learning algorithms MLP, RF, LR, J48, SVM, SMO, LWL, KStar, lazy-DT, ML-KNN, and KNN based on the hierarchical clustering method depicted in Figure 2. For KNN's combination function, we used simple unweighted voting for K=3, based on Euclidean distance. The comparison of MAE and R-squared is illustrated in Table 3.

## 5 EXPERIMENTS AND RESULTS

This section discusses the result of our experiments carried out to evaluate and compare the performance of MLP, RF, LR, J48, SVM, SMO, LWL, KStar, Lazy-DT, ML-KNN, and KNN algorithms. According to Table 3, KStar algorithm outperforms the other algorithms and shows the highest R-squared value (close to 1 or 100%) and lowest MAE compared to other algorithms tested in this work. Based on the overall measure of the fit of the model, we compare the best of eager methods with that of lazy methods. Among the eager methods tested, LR and J48 show the lowest MAE that is the highest accuracy, and LR shows the lowest variance (i.e., highest R-squared), while KNN and KStar show the minimum variance (i.e., highest R-squared value) among the lazy methods. In addition, KStar shows the highest accuracy (i.e., lowest MAE) compared to other ML methods tested in this work. In fact, KStar algorithm decreased the prediction error by 69%, 41%, and 8% for single-family, town-house, and condo respectively, compared to LR method. KStar decreased the prediction error by 71%, 55%, and 5% for single-family, town-house, and condo respectively, compared to RF method. Also, KStar decreased the prediction error by 67%, 55%, and 31% for single-family, town-house, and condo respectively, compared to J48 method. Finally, KStar decreased the prediction error by 71%, 49%, and 6.8% for single-family, town-house, and condo respectively, compared to KNN method. Although KStar beats all eager methods in terms of prediction accuracy, this is not necessarily found to be an overarching trend while comparing the remaining lazy vs. eager methods in this work. For instance, LR and J48 show lower variance on average, compared to lazy-DT. Based on Table 3, the low variations of MAE measure across different algorithms for the town-house records compared to single-family and condo is happening due to skewness of the data set: the number of single-family properties, and then condo, dominate the data set compared to town-house records, which indicates the skewness of the data set.Figure 9 illustrates the regression model outputs vs. the measured rent price. The results show that KStar regression model provides the best fit. However, not all the lazy methods discussed in this work dominate all the other eager methods in terms of variance and/or accuracy. All algorithms perform relatively well when it comes to the town-house data. We observed that for town-house records, the performance of the learning method LR is very close to KNN lazy learning method.

## 6 CONCLUSION

Although eager methods require less memory space compared to lazy algorithms, it is not considered an outstanding advantage in the real-estate housing domain. After all, we are not dealing with streaming data. Based on our experiments, real estate rent prices are not subject to change within periods shorter than 6 months, hence we don't consider this a real-time problem. Therefore, even though lazy learning methods show to be slower in terms of execution time compared to eager methods, we are still fine with lazy methods due to their high prediction accuracy and low variance. Based

Table 3: Comparison of eager vs. lazy learners for rent prediction using VA housing data set. The values show the average evaluation measures $R^2$ and MAE. Higher R-squared (R2) values show lower variance, and lower Mean Absolute Error (MAE) shows higher accuracy.

| Algorithm | Single-Family | | Town-House | | Condo | |
|---|---|---|---|---|---|---|
| | $R^2$ | MAE | $R^2$ | MAE | $R^2$ | MAE |
| MLP | 0.58 | 410 | 0.91 | 105.4 | 0.88 | 152.4 |
| RF | 0.68 | 322.7 | 0.78 | 109.7 | 0.90 | 109.3 |
| LR | 0.79 | 294.1 | 0.96 | 83.37 | 0.90 | 112.7 |
| J48 | 0.70 | 280.7 | 0.87 | 110.48 | 0.90 | 150.48 |
| SVM | 0.60 | 300.2 | 0.84 | 120 | 0.90 | 101.2 |
| SMO | 0.70 | 342.02 | 0.90 | 98.02 | 0.90 | 254.1 |
| LWL | 0.86 | 299.2 | 0.95 | 98.1 | 0.90 | 121.4 |
| KStar | 0.95 | 91.7 | 0.97 | 49.3 | 0.90 | 103.2 |
| Lazy-DT | 0.81 | 399.6 | 0.89 | 89.6 | 0.90 | 121.6 |
| ML_KNN | 0.82 | 289.6 | 0.80 | 100.6 | 0.90 | 108.6 |
| KNN | 0.93 | 321.065 | 0.92 | 97.15 | 0.90 | 110.78 |

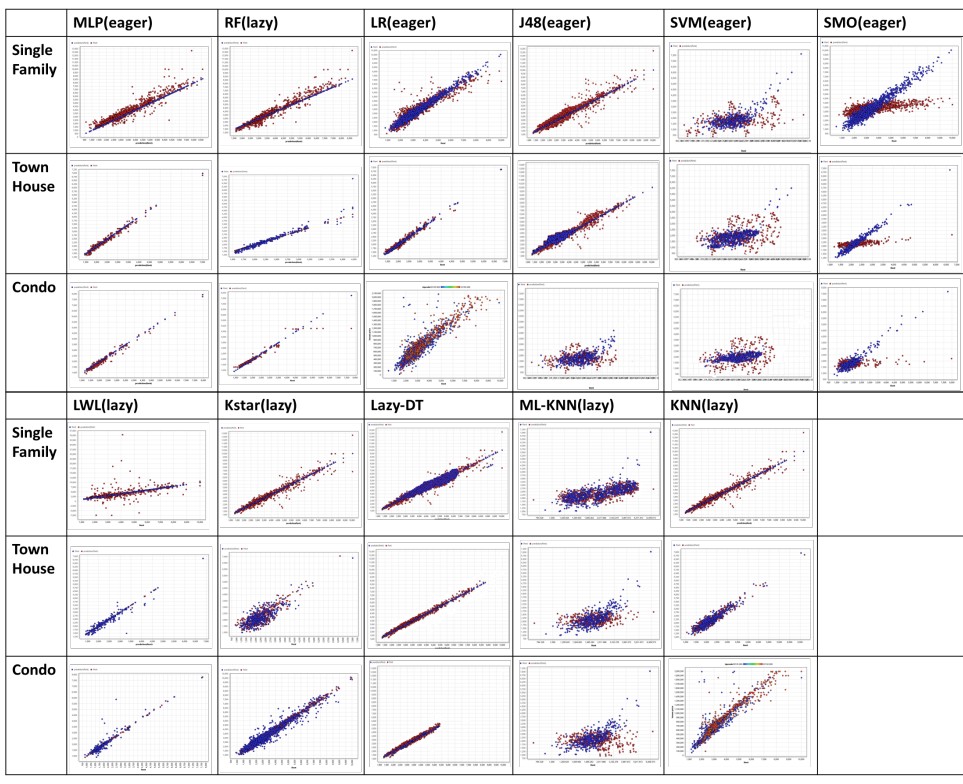

Figure 9: Predicted rent (blue) vs. the measured rent (red).

on this study, the fact that KStar algorithm outperformed the other techniques shows a unique performance in dealing with noisy and imbalanced housing dataset attributes in comparison with other Lazy learners. Furthermore, compared to other eager decision tree counterparts in this work (like RF), J48 shows a better performance in dealing with the sparsity of data by delaying the construction of decision tree until the last step, and using known attributes to induce an accurate prediction. In addition, LR and J48 algorithms show lower variance on average, compared to lazy-DT, which indicates that not all the lazy methods discussed in this work dominate all the other eager methods in terms of variance and/or accuracy.

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
