# OpenReview forum: "An Exhaustive Analysis of Lazy vs. Eager Learning Methods for Real-Estate Property Investment"
_ICLR.cc/2019/Conference_

### Official Review · AnonReviewer1 · 2018-11-01
**Empirical investigation with flaws and no novelty.**

**Rating:** 2
**Confidence:** 4

**Review:**

The paper investigates different machine learning approaches to model and
predict the return on property investments, in particular with respect to eager
and lazy learning techniques. The authors evaluate those different techniques on
a dataset of properties in Virginia. They conclude that lazy techniques provide
better performance than eager ones.

The paper is a purely empirical study that does not introduce any novel
machine learning or evaluation techniques. The authors use the off-the-shelf
WEKA toolbox. The results are not clear, given that only a single data set was
used to evaluate the different approaches, and general recommendations cannot be
made.

The paper is not well written and the descriptions do not convey what the
authors have done very well. An example of this is Figure 3, which purports to
show the average rent (or rent distribution?) for different housing types. There
are multiple categories in there that are not valid housing types ("Make Me
Move") and the rents shown are incorrect (e.g. more than a million for a town
house). It is also unclear why the average rent for a single family house is
approximately 4 times as much as for a town house.

The problems with Figure 3 are exemplary of the paper; the other issues are too
numerous to list.

In summary, this paper should be rejected.

---

### Official Review · AnonReviewer2 · 2018-11-02
**Concerns about methodological contributions**

**Rating:** 4
**Confidence:** 4

**Review:**

Quality: your paper provides a comprehensive evaluation of a number of methods. However, I am not convinced by the choice of methods or the need for them to be compared as such.

Clarity: the paper is unclear in a number of places.

Originality: as far as I am aware, the paper is novel.

Significance: As stated above, I do not believe you have made the case that we need a comparison of these methods on this dataset.

More detailed comments:

1. The distinction between lazy and eager learning is not one that I find particularly helpful. Perhaps this is my own ignorance. I checked two standard ML textbooks (Bishop, Murphy) and neither mentions this distinction. As the crux of your paper is a comparison between these methods, you need to be very upfront about why this dichotomy is a useful one. And as your paper is very application-oriented, the justification needs to be related to that application.

2. Your data is, fundamentally, spatial (actually spatiotemporal). Thus it would be appropriate to consider models that specifically account for spatial structure, e.g.  Gaussian processes.

3. Do you have a novel methodological or theoretical contribution? A comparison of methods on an important dataset is no doubt useful, but I don't think ICLR is the right venue.

---

### Official Review · AnonReviewer3 · 2018-11-07
**An application paper with no novelty and probably not suitable for the venue**

**Rating:** 3
**Confidence:** 5

**Review:**

The authors compare a collection of machine learning models to predict the expected rental income from an investment property. The dataset they use to train their model is fairly small (around 4K transactions). In addition to using house specific features the authors use other macro features, such as, walk score etc. Using this the authors compare a set of machine learning models and report their findings.

While the work presented in the paper is informative, I feel there are a number of issues with the paper, making it unsuitable for publication in ICLR. Some of them include:
- There is no new novel model or technique proposed in the paper. It is essentially a collection of experiments run on some dataset with reported findings.
- The dataset used is really small. Making the generalizability of the results somewhat questionable.
- Lastly, even though there is nothing technically wrong with the presented work, I feel that ICLR is not the best venue for such works. Perhaps a data science conference, such as, KDD/WSDM might be much better suited.

---

### Meta-Review · Area_Chair1 · 2018-12-01
**Lack of novelty and potentially flawed empirical study**

**Confidence:** 5
**Recommendation:** Reject

**Metareview:**

The paper evaluates several off-the-shelf algorithms for predicting the return on real-estate property investment. The problem is interesting, but there is a consensus that the paper contains little technical novelty, and the empirical study on a fairly small dataset is also not convincing.